# Regularizing Deep Neural Networks with Stochastic Estimators of Hessian Trace

## Abstract

In this paper we develop a novel regularization method for deep neural networks by penalizing the trace of Hessian. This regularizer is motivated by a recent guarantee bound of the generalization error. Hutchinson method is a classical unbiased estimator for the trace of a matrix, but it is very time-consuming on deep learning models. Hence a dropout scheme is proposed to efficiently implements the Hutchinson method. Then we discuss a connection to linear stability of a nonlinear dynamical system. Experiments demonstrate that our method outperforms existing regularizers such as Jacobian, confidence penalty, and label smoothing. Our regularization method is also orthogonal to data augmentation methods, achieving the best performance when our method is combined with data augmentation.

## 1 Introduction

Deep neural networks (DNNs) are developing rapidly and are widely used in many fields such as image classification, machine translation, language modeling and speech recognition. As more and more models are proposed in the literature, deep neural networks have shown remarkable improvements in performance. However, among various learning problems, over-fitting on training data is a great problem that affects the test accruacy. So a certain regularization method is often needed in the training process.

In linear models, Ridge Regression (Hoerl & Kennard, 1970) and Lasso (Tibshirani, 1996) are usually used to avoid over-fitting. They are also called $L_2$ and $L_1$ regularization. $L_2$ regularization has the effect of shrinkage while $L_1$ regularization can be conductive to both shrinkage and sparsity. From the Bayesian perspective, $L_2$ and $L_1$ regularization can be interpreted with normal prior distribution and laplace prior distribution respectively.

Apart from $L_2$ and $L_1$ regularization, there are many other forms of regularizers in DNNs. The most widely used one is Weight-Decay (Krogh & Hertz, 1992). Loshchilov & Hutter (2019) also showed that $L_2$ regularization and Weight-Decay are not identical. Dropout (Srivastava et al., 2014) is another method to avoid over-fitting by reducing co-adapting between units in neural networks. Dropout has inspired a large body of work studying its effects (Wager et al. (2013); Helmbold & Long (2015); Wei et al. (2020)). After dropout, various regularization schemes can be applied additionally.

In this paper, we propose a new regularization by penalizing the trace of second derivative of loss function. We refer to our regularization method as Stochastic Estimators of Hessian Trace (SEHT). On one hand, our hessian regularization is valuable to guarantee good generalization. On the other hand, from the perspective of dynamical system, it influences the stability of the system, in which parameters move in the parameter space on the basis of training data. In our experiments, Hessian regularization shows competing test performance and low time consumption with our stochastic algorithm.

## 2 Related Work

There are many regularization methods in previous work. Label Smoothing (Szegedy et al., 2016) estimates the marginalized effect of label-dropout and reduces over-fitting by preventing a network from assigning full probability to each training example. Confidence Penalty (Pereyra et al., 2017)

prevents peaked distributions, leading to better generalization. A network appears to be overconfident when it places all probability on a single class in the training set, which is often a symptom of over-fitting. DropBlock (Ghiasi et al., 2018) is a structured form of dropout, it drops contiguous regions from a feature map of a layer instead of dropping out independent random units.

Data augmentation methods are also used in practice to improve model's accuracy and robustness when training neural networks. Cutout (DeVries & Taylor, 2017) is a data augmentation method where parts of the input examples are zeroed out, in order to encourage the network to focus more on less prominent features, then generalize to situations like occlusion. Mixup (Zhang et al., 2017) extends the training distribution by incorporating the prior knowledge that linear interpolations of feature vectors should lead to linear interpolations of the associated targets.

Sokolić et al. (2017) first proposed Jacobian regularization, a method focusing on the norm of Jacobian matrix with respect to input data. It was proved that generalization error can be bounded by the norm of Jacobian matrix. Besides that, Jacobian matrix shows improved stability of the model predictions against input perturbations according to Taylor expansion. Hoffman et al. (2019) showed that Jacobian regularization enlarges the size of decision cells and is practically effective in improving the generalization error and robustness of the models. To simplify calculation, stochastic algorithm of Jacobian regularization was also proposed.

Motivated by Jacobian regularization, we consider the generalization error and stability of the model respect to Hessian matrix. Then we combine Linear Stability Analysis and propose Hessian regularization with corresponding stochastic algorithms. We compare our Hessian regularization with other methods and demonstrate promising performance in experiments. The main idea to estimate the trace of Hessian matrix is Hutchinson Method(Avron & Toledo, 2011) and the algorithm was also discussed by Yao et al. (2020a). We make an improvement by designing a new probability distribution to dropout parameters which decrease time consumption obviously without losing generalization.

Hessian information is powerful tool used on analyzing the property of neural networks. Yao et al. (2020b) designed AdaHessian, a second order stochastic optimization algorithm. Yu et al. (2021) used Hessian trace to measure sensitivity, developing a Hessian Aware Pruning method to find insensitive parameters in a neural network model and a Neural Implant technique to alleviate accuracy degradation. However, their methods are static in essence, we focus on dynamical motion of parameters in parameter space. Sankar et al. (2021) also proposed a Hessian regularization. They forcused on the layerwise loss landscape via the eigenspectrum of the Hessian at each layer. We start from different perspectives, generalization error and dynamical system of parameters. Our experiments also shows better results than Sankar et al. (2021)'s method.

## 3 HESSIAN REGULARIZATION

Here we introduce a Hessian regularization method based on generalization error and corresponding stochastic algorithms in details. Then we discuss linear stability analysis to explain why Hessian regularization can prevent neural networks from over-fitting.

### 3.1 TRACE OF HESSIAN MATRIX

In this study, we consider a multi-class classification problem. Input $x$ is a N-dimensional vector, where $x \in \mathcal{X} \subseteq \mathbb{R}^N$, with $\mathcal{X}$ is the input space. $\mathcal{Y} = \{1, 2, ..., M\}$ is the label space, which means that we have M classes. Each input $x$ has a label $y \in \mathcal{Y}$. Sample space is defined as $\mathcal{S} = \mathcal{X} \times \mathcal{Y}$. An element of $\mathcal{S}$ is denoted by $s = (x, y)$. We assume that samples $s$ from $S$ are drawn according to a probability distribution $P$. A training set of n samples drawn from $P$ is denoted by $S_n = \{s_i\}_{i=1}^n = \{(x_i, y_i)\}_{i=1}^n$.

Our goal is to find a classification function $f$, which takes $x \in \mathbb{R}^N$ as input and outputs $z = f(x)$. $z$ is a M-dimensional score vector, where each element $z_i$ is the score $x$ belonging to category $i \in \mathcal{Y}$. The highest score indicates the most probable label. So the estimated label is given as $\hat{y} = g(x) = \arg\max_i z_i = \arg\max_{i \in \mathcal{Y}} f(x)_i$.

A loss function is used to measure the discrepancy between the true label $\boldsymbol{y}$ and the estimated label $g(\boldsymbol{x})$. In this paper, we use the cross-entropy loss,

$$\ell(f(x), y) = -\log \frac{\exp(f(\boldsymbol{x})_{\boldsymbol{y}})}{\sum_{\boldsymbol{y}'} \exp(f(\boldsymbol{x})_{\boldsymbol{y}'})}. \tag{1}$$

The empirical loss of the classifier $f(\boldsymbol{x})$ associated with the training set is defined as

$$\ell_{emp}(f) = \hat{\mathbb{E}}[\ell(f(\boldsymbol{x}), \boldsymbol{y})] = \frac{1}{n} \sum_{\boldsymbol{s}_i \in \mathcal{S}_n} \ell(f(\boldsymbol{x}_i), \boldsymbol{y}_i), \tag{2}$$

and the expected loss of the classifier $f(\boldsymbol{x})$ is defined as

$$\ell_{exp}(f) = \mathbb{E}[\ell(f(\boldsymbol{x}), \boldsymbol{y})] = \mathbb{E}_{\boldsymbol{s} \sim P}[\ell(f(\boldsymbol{x}), \boldsymbol{y})], \tag{3}$$

then the difference between $\ell_{emp}(f)$ and $\ell_{exp}(f)$ is called generalization error:

$$GE(f) = ||\ell_{exp}(f) - \ell_{emp}(f)||. \tag{4}$$

Wei et al. (2020) showed generalization bound of linear models with cross-entropy loss of M classes. Let $\boldsymbol{W}$ is the weight matrix, $\mu(\boldsymbol{W}) = \hat{E}[\|\boldsymbol{J}\|_2]$ and $v(\boldsymbol{W}) = \hat{E}[tr(\boldsymbol{H})]$. For linear models, the Jacobian matrix $\boldsymbol{J}$ is a vector defined as $\frac{\partial \ell}{\partial \boldsymbol{z}}$ and the Hessian matrix $\boldsymbol{H}$ is defined as $\{\frac{\partial^2 \ell}{\partial \boldsymbol{z}_i \partial \boldsymbol{z}_j}\}_{M \times M}$. With probability $1 - \delta$ over the training examples, for all weight matrices $\boldsymbol{W}$ satisfying the norm bound $||\boldsymbol{W}^T||_{2,1} \leq A$, the following bound holds:

$$\mathbb{E}[\bar{\ell}] - 1.01\hat{\mathbb{E}}[\bar{\ell}] \lesssim \frac{(A\mu(\boldsymbol{W}))^{\frac{2}{3}}(\theta B)^{\frac{1}{3}}}{n^{\frac{1}{3}}} + \frac{A\sqrt{Bv(\boldsymbol{W})\theta}}{\sqrt{n}} + \frac{BA^2\theta}{n(\log^2(\frac{BA^2\theta}{v(\boldsymbol{W})n}) + 1)} + \zeta. \tag{5}$$

Here with some fixed bound $B > 0$,

$$\bar{\ell} = min\{\ell, B\},$$
$$\|\boldsymbol{W}\|_{2,1} = \sum_j \sqrt{\sum_i (W_{ij}^2)},$$
$$\theta = \log^3(nM) \max_i \|x_i\|_2^2, \tag{6}$$
$$\zeta = \frac{B(\log(1/\delta) + \log\log n)}{n}.$$

So one can guarantee good generalization when the trace of Hessian matrix and norm of Jacobian matrix are small. On one hand, when learning with gradient descent, we want to find a local or global minimum of loss function. Naturally, at minimum the gradient is zero and the norm of Jacobian matrix is small near minimum. So gradient descent helps us to ensure the norm of Jacobian Matrix to be small.

On the other hand, the trace of Hessian Matrix is hard to be constrained by gradient descent. From this aspect, we proposed Hessian regularization for linear models as:

$$\frac{1}{n}tr(\boldsymbol{H}_{\ell,\boldsymbol{z}}). \tag{7}$$

It's the trace of second derivative of loss $\ell$ with respect to output of linear model $\boldsymbol{z}$, which is also the end nodes of a linear model. A DNN is consist of many layers, with each layer being viewed as a linear model (except the nonlinear activation functions). Thus. we generalize the Hessian regularization to every node in a DNN and define it as

$$\frac{1}{n}tr(\boldsymbol{H}_{\ell,\omega}). \tag{8}$$

It's the trace of second derivative of loss $\ell$ with respect to parameters $\omega$.

Here we define a new loss with our Hessian regularization as

$$Loss = \ell_{emp}(f) + \lambda \cdot \frac{1}{n}tr(\boldsymbol{H}_{\ell,\omega}), \tag{9}$$

where $\lambda$ controls the strength of our Hessian regularization.

## 3.2 HUTCHINSON METHOD

In a typical DNN, there are more than millions of parameters. So the calculation of Hessian matrix is difficult. Hutchinson Method (Avron & Toledo, 2011) is an unbiased estimator for the trace of a matrix. Let $A$ be an $n \times n$ symmetric matrix with $tr(A) \neq 0$. Let $\sigma$ be a random vector whose entries are i.i.d Rademacher random variables ($Pr(\sigma_i = \pm 1) = \frac{1}{2}$), then $\sigma^T A \sigma$ is an unbiased estimator of $tr(A)$, based on the following equation:

$$tr(A) = tr(AI) = tr(A\mathbb{E}[\sigma\sigma^T]) = \mathbb{E}[tr(A\sigma\sigma^T)] = \mathbb{E}[\sigma^T A \sigma]. \tag{10}$$

In this paper, we consider the trace of Hessian matrix $H$, which is the second derivative matrix. Since the Rademacher random vector is irrelevant to network parameters, we expand the expression of Hutchinson estimator as follow:

$$\sigma^T H \sigma = \sigma^T \frac{d\frac{dl}{d\omega}}{d\omega}\sigma = \sigma^T\left(\frac{d\frac{dl}{d\omega}}{d\omega} \cdot \sigma + \frac{dl}{d\omega} \cdot \frac{d\sigma}{d\omega}\right) = \sigma^T\left(\frac{d(\frac{dl}{d\omega} \cdot \sigma)}{d\omega}\right). \tag{11}$$

Based on Equation 11 and $\frac{d\sigma}{d\omega} = 0$, we only have to calculate the gradient $g = \frac{dl}{d\omega}$ and the derivative of $\frac{dl}{d\omega} \cdot \sigma$, instead of the whole Hessian matrix. The whole calculation process only include two inner products and two derivations. We refer to the Hutchinson stochastic estimator of Hessian trace as SEHT-H.

---

**Algorithm 1:** SEHT-H

**Input:** n-dimensional gradient $g$
**Output:** Estimation of $tr(H)$

1   $trace = 0$ ;
2   **for** $i = 1$ **to** $maxIter$ **do**
3     $\sigma \sim Rademacher(n)$ ;     /* n-dim vector with each element sampled from Rademacher distribution. */
4     $v = g \cdot \sigma$ ;                         /* inner product */
5     $h = \frac{dv}{d\omega}$ ;                       /* derivative of $v$ */
6     $t = \sigma^T \cdot h$ ;                     /* inner product */
7     $trace \mathrel{+}= t$ ;
8   **end**
9   **return** $\frac{trace}{maxIter}$

---

Even though it's a stochastic algorithm, it cost much time because of a great number of parameters. So we propose another efficient algorithm below based on the basic idea of Hutchinson Method.

## 3.3 DROPOUT METHOD

This stochastic algorithm is inspired by Dropout (Srivastava et al., 2014). Every node in the neural network has a probability $p$ to be igonred in the training process to reduce co-adaptations. In our Hessian regularization, we want to lower the trace $tr(H) = \sum_i \frac{\partial^2 \ell}{\partial \omega_i \partial \omega_i}$, the sum of diagonal elements of Hessian matrix. Based on the idea of Dropout, we can ignore some parameters in constraining the $tr(H)$, since reducing the partial sum of diagonal elements can have a large chance to reduce the total sum. The partial sum of diagonal element is denoted as $\tilde{tr}(H)$. Considering the layer structures of neural networks, the process of randomized parameter selection can be divided into two steps: randomly select layers in neural network with probability $p_1$ and randomly select parameters in the selected layers with probability $p_2$. In other words, when carrying out Hessian regularization, we ignore other layers with probability $1 - p_1$, and ignore other parameters in selected layers with probability $1 - p_2$. In our experiment, we simply set $p_1 = p_2$.

In addition, Hutchinson method shows a technique, which avoids to calculate the whole Hessian matrix. Here we define a new probability distribution $Q(p)$ (if $x \sim Q(p)$, then $Pr(x = \pm 1) = p$ and $Pr(x = 0) = 1 - 2p$). Then, let $\sigma$ be a random vector whose entries are i.i.d $Q$ random variables,

$$\mathbb{E}[\sigma\sigma^T | fix\ the\ position\ of\ 0\ in\ \sigma] = \tilde{I}. \tag{12}$$

Here $\tilde{I} = \mathrm{diag}(0, 1)$, a diagonal matrix with diagonal elements equal to 0 or 1.

Similar to Equation 10, if we fix the position of 0 in $\sigma$, we have unbiased estimator of the partial sum of diagonal elements:

$$\tilde{tr}(\boldsymbol{A}) = tr(\boldsymbol{A}\tilde{I}) = tr(\boldsymbol{A}\mathbb{E}[\sigma\sigma^T]) = \mathbb{E}[tr(\boldsymbol{A}\sigma\sigma^T)] = \mathbb{E}[\sigma^T\boldsymbol{A}\sigma]. \tag{13}$$

We can expand the expression same as Equation 11 and transform the calculation process into two inner product and two derivation. We name this method as SEHT-D.

---

**Algorithm 2:** SEHT-D

---

**Input:** probability p, parameter $\omega$ in selected layers, and corresponding n-dim gradient $\boldsymbol{g}$
**Output:** estimation of $\tilde{tr}(\boldsymbol{H})$

1   $trace = 0$ ;
2   **for** $i = 1$ **to** $maxIter$ **do**
3      $\sigma \sim Q(p)$;     /* n-dim vector with each element sampled from $Q(p)$ distribution */
4      $\boldsymbol{v} = \boldsymbol{g} \cdot \sigma$ ;                                  /* inner product */
5      $\boldsymbol{h} = \frac{d\boldsymbol{v}}{d\omega}$ ;                                /* derivative of $\boldsymbol{v}$ */
6      $\boldsymbol{t} = \sigma^T \cdot \boldsymbol{h}$ ;                              /* inner product */
7      $trace \mathrel{+}= \boldsymbol{t}$ ;
8   **end**
9   **return** $\frac{trace}{maxIter}$

---

In our experiments, we mainly test the performance of this method. Compared with other regularization methods, our Hessian regularization shows improved test performance with fast training speed.

### 3.4 LINEAR STABILITY ANALYSIS

Training process can be regarded as a motion in the parameter space, from the initial parameter to a local or global minimum. Current parameters is a point in the parameter space and gradient descent is the move of the parameter point each time. Then we can see gradient descent from another perspective. Original gradient descent is defined as a series of discrete updates:

$$\omega_{t+1} = \omega_t - \eta g_t. \tag{14}$$

Here $\omega_t$ is the parameters in step t, $\eta$ is learning rate and $g_t$ is gradient.

If we consider learning rate as discrete time interval to move $\omega$ in parameter space, $\Delta t = \eta$, then

$$\frac{\Delta\omega}{\Delta t} = -g(\omega, x). \tag{15}$$

We assume time interval or learning rate is small enough, approximately we get a contiunous form:

$$\frac{d\omega}{dt} = -g(\omega, x). \tag{16}$$

Thereafter, with an initial condition, we have the complete trajectory of parameter point based on ordinary differential equation (ODE) theory. The process of gradient descent is transformed to a Nonlinear Dynamical System. So we introduce Linear Stability Analysis of Nonlinear Dynamical Systems below.

Nonlinear Dynamical System (Thomas Witelski & Mark Bowen, 2015) is defined as a differential function $\frac{d\boldsymbol{x}}{dt} = \boldsymbol{f}(\boldsymbol{x}, t)$ with an initial condition $\boldsymbol{x}(0) = \boldsymbol{x}_0$. $\boldsymbol{x}(t) = (x_1(t), x_2(t), \ldots, x_n(t)) \in \mathbb{R}^n$ is the vector of state variables, describes $n$ properties of interest in the system, evolving for times $t \geq 0$ and starting from a given initial state $\boldsymbol{x}(0) = \boldsymbol{x}_0$. The rate functions for the rates of change of each $x_i$, $\frac{dx_i}{dt} = f_i$, have similarly been collected in a vector $\boldsymbol{f} = (f_1, f_2, \ldots, f_n)$, where each $f_i$ can potentially depend on all of the state variables. Since $f_i$ doesn't have to be a linear function, the system is called Nonlinear Dynamical System. A classic example of a dynamical system from mechanics is the system for motion of a particle. A system is called non-autonomous if the rate function have an explicit dependence on time. In this paper, we only focus on autonomous systems,

$$\frac{d\boldsymbol{x}}{dt} = \boldsymbol{f}(\boldsymbol{x}), \ \boldsymbol{x}(0) = \boldsymbol{x}_0. \tag{17}$$

The equilibrium points $x_*$ are defined by positions where rate functions vanish, $f(x_*) = 0$. If any solution starting near an equilibrium point leaves the neighbourhood of $x_*$ as $t \to \infty$, then $x_*$ is called asymptotically unstable, while if all solutions starting within the neighbourhood approach $x_*$ as $t \to \infty$ then the equilibrium is called asymptotically stable. Lyapunov (1992) gave more rigorous definition and discussion, known as Lyapunov Stability Theory. The idea of Lyapunov Stability can be extended to infinite-dimensional manifolds, where it is known as Structural Stability (Pugh & Peixoto, 2008), which concerns the behavior of different but "nearby" solutions to differential equations.

For a basic Nonlinear Dynamical System, to ensure equilibrium point $x_*$ is stable, we need to construct Jacobian matrix $J = \frac{\partial f}{\partial x}$. Given the equilibrium point $x_*$, $J(x_*)$ is a constant matrix. Using the conclusion from Linear State Space Model, if all eigenvalues of $J(x_*)$ have real parts less than zero, then $x_*$ is stable. If at least one of the eigenvalues has a real part greater than zero, then $x_*$ is unstable.

Back to gradient descent, it can be regarded as a Nonlinear Dynamical System, according to Equation 16. The local or global minimum, which is the goal of gradient descent, is the equilibrium point in such system, since that minimum point satisfies the condition $\frac{d\omega}{dt} = 0$. The Jacobian matrix in this dynamical system is the negative of the Hessian matrix in our Hessian regularization, $J(\omega_*) = \frac{-\partial g}{\partial \omega} = \frac{-\partial \frac{\partial \ell}{\partial \omega}}{\partial \omega} = -H_\ell$. And the trace and the eigenvalues of $J(\omega_*)$ are also negative of the trace and the eigenvalues of $H_\ell$.

It's easy to see that in our Hessian regularization, we lower the trace of Hessian Matrix $H_\ell$, thus increasing the trace of the Jacobian matrix $J(\omega_*)$. However, in real matrix, complex eigenvalues are always conjugate and the trace are always real number. When we increase the trace, we increase the real parts of eigenvalues of $J(\omega_*)$ to some extent. In other word, the goal is to preclude the stability of equilibrium point by our Hessian regularization.

Why we want instability in this dynamical system of gradient descent? The stability of local or global minimum shows the stability toward training data. The whole dynamical system of gradient descent is a motion based on training set since the motion of parameters is decided by Equation 17. Equation 17 is consist of two parts: one is the initial condition, which is randomized in a DNN, while the other part is the differential function Equation 16, which is directly determined by training data. In other words, with different training data, the parameters have different trajectory in parameter space. It only use information about training data and the equilibrium point depends on training data. Therefore, the stability is relevant to training set and reducing the stability to some extent can avoid over-fitting to training data.

## 4 EXPERIMENTS

We evaluate our Hessian regularization with other regularization methods on a variety of datasets. We also combine data augmentation methods with our Hessian regularization to test its efficacy.

### 4.1 IMAGE CLASSIFICATION

#### 4.1.1 CIFAR-10

The CIFAR-10 dataset consists of 60000 32x32 colour images in 10 classes, with 6000 images per class. There are 50000 training images and 10000 test images.

For our experiment, we use ResNet-18 (He et al., 2016) as the backbone neural network. For all models, we use Weight Decay of $5 \times 10^{-4}$. We set learning rate 0.01, batch size 32, momentum 0.9 and all models were trained 200 epochs with Cosine Annealing (Loshchilov & Hutter, 2016). For Jacobian regularization, we set number of projections $n_{proj} = 1$ and weight values $\lambda_{JR} = 0.01$. For DropBlock, $block\_size = 7$ and $keep\_prob = 0.9$. For Confidence Penalty and Label Smoothing, we set weight value 0.1, which is found to work best on CIFAR-10 by Pereyra et al. (2017) in their experiments. For our Hessian regularization SEHT-D, we also set weight value 0.1, testing with probability value 0.01 and 0.05. In addition, we always add output layer in the selected layer, which corresponds to Equation 7. Cutout size of $16 \times 16$ pixels is used in our experiment, based on the

validation results mentioned by DeVries & Taylor (2017). For mixup, $\alpha = 0.1$. Averages and $95\%$ confidence intervals are estimated over 5 distinct runs.

We also observe the computational cost of the training with SEHT-D (maxIter=1, prob=0.01) to be only 1.2 times that of the baseline. Although increasing the probability to select parameters can improve test accuracy, the time consumption will increase a lot. In our experiment, SEHT-D (maxIter=1, prob=0.05) costs 1.5 times of the baseline.

In our experiments, we find that Jacobian regularization and Dropblock have worse performance than the baseline with Weight-Decay. Confidence Penalty has slight improvement and Label Smoothing has obvious improvement. However, our SEHT-D shows better results, compared with all other regularization methods which are tested. In addition, our SEHT is suitable for combination with other methods. When we combin SEHT-D with mixup, we get the best test accuracy in our experiment, 1.45 more than baseline.

In Sankar et al. (2021)'s experiment, they got test accuracy 88.13 on CIFAR-10 with ResNet-18, which is much worse than our result: 94.35 with SEHT-D (maxIter=1, prob=0.01) and 94.37 with SEHT-D (maxIter=1, prob=0.05). Moreover, their improvement based on their methods is only 0.02 for full-network method and 0.10 for middle-network method. Our Hessian regularization method improves the model 0.35 on test accuracy with SEHT-D (maxIter=1, prob=0.01) and improves 0.37 on test accuracy with SEHT-D (maxIter=1, prob=0.05), which is over 3 times of their improvement.

Table 1: ResNet-18 on CIFAR-10

| Model | Test Accuracy |
|---|---|
| Baseline with Weight-Decay | $94.00 \pm 0.47$ |
| Jacobian | $89.23 \pm 1.02$ |
| DropBlock | $89.23 \pm 0.44$ |
| Sanker's Method for Full Network | $88.05 \pm 0.22$ |
| Sanker's Method for Middle Network | $88.13 \pm 0.12$ |
| Confidence Penalty | $94.01 \pm 0.40$ |
| Label Smoothing | $94.26 \pm 0.26$ |
| **SEHT-D (maxIter=1, prob=0.01)** | $\mathbf{94.35 \pm 0.18}$ |
| **SEHT-D (maxIter=1, prob=0.05)** | $\mathbf{94.37 \pm 0.27}$ |
| **SEHT-D (maxIter=1, prob=0.01) + Label Smoothing** | $\mathbf{94.38 \pm 0.24}$ |
| cutout | $94.02 \pm 0.22$ |
| mixup | $95.39 \pm 0.13$ |
| **SEHT-D (maxIter=1, prob=0.01) + mixup** | $\mathbf{95.45 \pm 0.06}$ |

### 4.1.2 CIFAR-100

CIFAR-100 dataset is just like the CIFAR-10, except it has 100 classes containing 600 images each. There are 500 training images and 100 testing images per class.

We use Wide Residual Networks (WRN) (Zagoruyko & Komodakis, 2016) as the backbone neural network. We use WRN-28-10 specifically, with depth 28 and fixed widening factor of 10. For all models, we use Weight Decay of $5 \times 10^{-4}$. We set batch size 32, momentum 0.9 and all models were trained 200 epochs. The learning rate is initially set to 0.1 and is scheduled to decrease by a factor of 5 after each of the 60th, 120th, and 160th epochs. We test Dropout with a drop probability of $p = 0.3$, determined by Zagoruyko & Komodakis (2016)'s cross-validation. For Confidence Penalty, Label Smoothing and our Hessian regularization, we set weight value 0.1. Cutout size of 8 × 8 pixels is used according to DeVries & Taylor (2017)'s validation results. For mixup, we still set $\alpha = 0.1$. Averages and $95\%$ confidence intervals of top-1 accuracy and top-5 accuracy are estimated over 5 distinct runs.

In this experiments, Confidence Penalty has better top-5 accuracy, worse top-1 accuracy and Label Smoothing has better top-1 accuracy, worse top-5 accuracy, compared with baseline method. However, our SEHT-D method shows better results on both top-1 accuracy and top-5 accuracy, improving 1.13 and 0.65 respectively. Our Hessian regularization method also perform better when combined

with other method on this dataset. When testing together with Dropout, our SEHT-D has best accuracy, compared with Label Smoothing and Confidence Penalty. Our SEHT-D also improves cutout for 0.39 on top-1 accuracy, 0.26 on top-5 accuracy, and improves mixup 0.21 on top-1 accuracy, 0.09 on top-5 accuracy.

Table 2: WRN-28-10 on CIFAR-100

| Model | Top-1 Accuracy | Top-5 Accuracy |
|---|---|---|
| Baseline with Weight-Decay | $73.79 \pm 2.68$ | $92.01 \pm 1.32$ |
| Confidence Penalty | $73.46 \pm 1.21$ | $92.16 \pm 0.58$ |
| Label Smoothing | $74.15 \pm 0.92$ | $90.40 \pm 0.73$ |
| **SEHT-D(maxIter=1, prob=0.01)** | **$74.92 \pm 0.77$** | **$92.66 \pm 0.54$** |
| Confidence Penalty + Dropout | $74.80 \pm 0.91$ | $93.09 \pm 0.51$ |
| Label Smoothing + Dropout | $72.89 \pm 1.57$ | $90.43 \pm 0.97$ |
| **SEHT-D(maxIter=1, prob=0.01) + Dropout** | **$77.75 \pm 0.37$** | **$94.38 \pm 0.09$** |
| cutout | $76.70 \pm 0.79$ | $93.72 \pm 0.40$ |
| **SEHT-D(maxIter=1, prob=0.01) + cutout** | **$77.09 \pm 0.37$** | **$93.98 \pm 0.21$** |
| mixup | $78.38 \pm 0.31$ | $94.37 \pm 0.31$ |
| **SEHT-D(maxIter=1, prob=0.01) + mixup** | **$78.59 \pm 0.46$** | **$94.46 \pm 0.30$** |

Our experiments on Image Classification shows that our Hessian regularization method outperforms other regularization methods and can be efficiently combined with data augmentation methods.

## 4.2 LANGUAGE MODELING

### 4.2.1 WIKI-TEXT2

The Wiki-Text language modeling dataset is a collection of over 100 million tokens extracted from the set of verified Good and Featured articles on Wikipedia.

We use a 2-layer LSTM (Hochreiter & Schmidhuber, 1997). The size of word embeddings is 512 and the number of hidden units per layer is 512. We run every algorithm for 40 epochs, with batch size 20, gradient clipping 0.25, and Dropout ratio 0.5. We perform a grid search over Dropout ratios $\{0, 0.1, 0.2, 0.3, 0.4, 0.5\}$ and find 0.5 to work best. We tune the initial learning rate from $\{0.001, 0.01, 0.1, 0.5, 1, 10, 20, 40\}$ and decrease the learning rate by factor of 4 when the validation error saturates. We find initial learning rate 20 works best. Parameters are initialized from a uniform distribution $[-0.1, 0.1]$. For label smoothing, we perform a grid search over weight values $\{0.001, 0.005, 0.01, 0.05, 0.1\}$ and find 0.01 to work best. For the confidence penalty, we perform a grid search over weight values $\{0.001, 0.005, 0.01, 0.05, 0.1\}$ and find 0.01 to work best. For our Hessian regularization, we perform a grid search over weight values $\{0.001, 0.005, 0.01, 0.05, 0.1\}$, probability values $\{0.01, 0.05\}$. Weight 0.01 and probability 0.05 work best. Averages and 95% confidence intervals are estimated over 5 distinct runs.

In this experiments with LSTM, our SEHT-D has the best test perplexity and Label Smoothing shows best validation perplexity. SEHT-D improves the model 2.61 on test perplexity. Confidence Penalty performs only sightly better than the baseline method.

Table 3: LSTM on Wiki-Text2

| Model | Validation Perplexity | Test Perplexity |
|---|---|---|
| Baseline with Dropout | $101.82 \pm 0.32$ | $95.65 \pm 0.19$ |
| Confidence Penalty | $101.39 \pm 0.32$ | $95.57 \pm 0.11$ |
| Label Smoothing | **$99.58 \pm 0.11$** | $95.03 \pm 0.58$ |
| **SEHT-D** | $100.69 \pm 0.53$ | **$94.86 \pm 0.50$** |

We also tested with a 2-layer GRU (Cho et al., 2014). The size of word embeddings is 512 and the number of hidden units per layer is 512. We run every algorithm for 40 epochs, with batch size 20, gradient clipping 0.25 and Dropout ratio 0.3. We perform a grid search over Dropout ratios $\{0, 0.1, 0.2, 0.3, 0.4, 0.5\}$ and find 0.3 to work best. We tune the initial learning rate from $\{0.001, 0.01, 0.1, 0.5, 1, 10, 20, 40\}$ and decrease the learning rate by factor of 4 when the validation error saturates. We find initial learning rate 20 works best, same as LSTM. Parameters are initialized from a uniform distribution $[-0.1, 0.1]$. For label smoothing, we perform a grid search over weight values $\{0.001, 0.005, 0.01, 0.05, 0.1\}$ and find 0.05 to work best. For the confidence penalty, we perform a grid search over weight values $\{0.001, 0.005, 0.01, 0.05, 0.1\}$ and find 0.005 to work best. For our Hessian regularization, we perform a grid search over weight values $\{0.001, 0.005, 0.01, 0.05, 0.1\}$. Weight 0.001 works best. We set probability values 0.01. Averages and $95\%$ confidence intervals are estimated over 5 distinct runs.

Our Hessian regularization method has both the best validation perplexity and the best test perplexity, improving 2.83 and 2.61 respectively compared with baseline method. Confidence Penalty surpasses Label Smoothing with GRU model, compared with LSTM. Label Smoothing also show better results than baseline.

Table 4: GRU on Wiki-Text2

| Model | Validation Perplexity | Test Perplexity |
|---|---|---|
| Baseline with Dropout | $119.04 \pm 4.67$ | $111.64 \pm 3.67$ |
| Confidence Penalty | $116.40 \pm 0.17$ | $109.27 \pm 0.05$ |
| Label Smoothing | $117.47 \pm 0.48$ | $110.46 \pm 0.87$ |
| **SEHT-D** | $\mathbf{116.21 \pm 0.60}$ | $\mathbf{109.03 \pm 0.30}$ |

Our experiments on Language Modelling demonstrate that all these three regularization methods can improve models, while our SEHT-D is the best.

## 5 CONCLUSION

We propose a new regularization method named as Stochastic Estimators of Hessian Trace (SEHT). Our method is motivated by a guarantee bound that a lower trace of the Hessian can result in a lower generalization error. We also explained our method with dynamical system theory. Our experiment shows that SEHT-D yields promising test performance with fast training speed. SEHT-D also achieves better results when combined with data augmentation methods.

## 6 REPRODUCIBILITY STATEMENT

Our codes are available at here .

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
