# OpenReview forum: "Regularizing Deep Neural Networks with Stochastic Estimators of Hessian Trace"
_ICLR.cc/2022/Conference — ICLR 2022 Submitted_

### Official Review · Reviewer_CMzT · 2021-11-01

**Correctness:** 2
**Technical Novelty And Significance:** 2
**Empirical Novelty And Significance:** 2
**Recommendation:** 5
**Confidence:** 3

**Main Review:**

Strengths:
- The connection of Hessian spectrum and trace to generalization is an interesting and promising area of research and it is great to see results for a method regularizing the Hessian that seems to outperform other regularization methods
- The structure of the paper is clear and the paper is understandable

Weaknesses:
- The connection between theoretical justification and practical method is not always clear and the relevant details could be expanded:

My feeling is that the connection between using the Hessian / Jacobian with respect to the logits vs with respect to the parameters is not as straight forward as it is made out to be in the paper and could be explained better. For example the paper states: "Naturally, at the minimum the gradient is zero and the norm of the Jacobian matrix is small near minimum."

But if we have for e.g. a nonlinear classifier f(x) = W y(x) where W are the weights of the last layer then dl/df is what is meant as Jacobian in the bound but the gradient in optimization is with respect to parameters, e.g. dl(f(x))/dW = dl/df * df/dW = dl/df * y(x). Could there perhaps be a way for the gradient to be small if y(x) is small without dl/df being small? This question could be expanded upon.

Similarly, an explanation of the difference between the trace of the loss Hessian w.r.t. to logits vs w.r.t to the weights would benefit the paper. For example it is well-known that the Hessian w.r.t. to weights for a deep neural network can have negative eigenvalues but the Hessian w.r.t. to logits for a convex loss has only positive eigenvalues. Does this matter to the trace and associated bound somehow?

Regarding the linear stability argument the paper could be improved by empirically demonstrating that regularizing the Hessian trace leads to an optimum with a Hessian that has "less stability", perhaps by analyzing the eigenvalues at the optimum. Furthermore, is there a connection between the idea that increasing instability helps generalization to the literature that says that flat optima help generalization?

- There is no discussion of using the reverse-over-reverse autodiff method for computing the Hessian vector product instead of a forward-over-reverse autodiff method which can save memory and hence could also be faster on a GPU:

In principle you can get the directional derivative of the scalar g^T(omega) * sigma with respect to omega in the direction of sigma by evaluating

dg/domega * sigma = (g(omega + eps * sigma) - g(omega)) / eps

or similarly

d(g*sigma)/domega * sigma = (g(omega + eps * sigma) * sigma - g(omega) * sigma) / eps

at a factor two of the cost of evaluating only the gradient (gradient evaluated at omega). Alternatively, forward-mode autodiff can be used to the same effect and the dropout idea can still be applied by setting some of the sigma to zero and hence eliminating the cost further due to dropped nodes in the computational graph.

- The experimental section does not analyse the regularizer itself or the connection to theory of the Hessian trace regularization and is instead focused on just demonstrating beating baselines:

There is currently no analysis of the trade-off of the bias of the trace estimator by using more dropout vs additional compute cost, just the statement "Although increasing the probability to select parameters can improve test accuracy, the time consumption will increase a lot". The paper could be improved by adding an empiricial analysis of the accuracy of the trace estimator.

There is also no analysis of how trace accuracy impacts generalization performance and trace itself, e.g. you could add plots showing how the true Hessian trace changes throughout training.

It would be good to see the training and cross validation curves for different hyperparams of overlayed penalty terms (like Hessian trace term, L2 norm term etc.) to see how they impact training and how stable training is to the corresponding hyperparameters. It would be good to have an analysis of how much does Hessian trace term add on top of hyper-param tuned other regularization, e.g. for Jacobian regularization the paper says just "we set number of projections nproj = 1 and weight values λJR = 0.01" which does not necessarily suggest that there was a sufficient hyperparameter tuning effort.

Another interesting question to analyze would be: How does the added regularizer impact the training dynamics, e.g. faster or slower convergence in the beginning vs end of training and why?


Some more general recommendations to improve the paper:
- Checking grammar and spelling typos will improve readability and quality of the paper
- State the generalization bound as a theorem as in the original paper (Theorem 4.1)
- In related work perhaps also connect to other works analyzing Hessian and Fisher matrix spectrum and related norms for generalization (e.g. "Fisher-Rao Metric, Geometry, and Complexity of Neural Networks")


**Summary Of The Paper:**

The authors propose a method for estimating the trace of the Hessian of the cross-entropy loss with respect to the weights of a neural network classifier. They suggest adding the trace estimator as a regularizing penalty term for training neural networks with improved generalization.

The authors give two theoretical motivations for regularizing the Hessian trace:
- For linear models the trace of the Hessian of the cross-entropy loss with respect to the logits appears as a factor in a term for bounding generalization error. The authors draw a connection between the Hessian with respect to the weights and the Hessian with respect to the logits and argue that penalizing the trace of the Hessian with respect to the weights also leads to a tighter bound on the generalization error.
- For the continuous gradient dynamics around the optimum the Hessian eigenspectrum is the negative eigenspectrum of the Jacobian of the gradient dynamics which determine the stability of the dynamic system. The authors argue that decreasing Hessian trace increases Jacobian trace and hence reduces data dependend stability which will avoid overfitting.

The authors' suggested method for estimating the trace uses the stochastic Hutchinson trace estimator which samples a random vector sigma with E(sigma sigma^T) = I to get E(sigma^T H sigma) = tr(H). The authors appear to use back-propagation through a back-propagation gradient approach to compute sigma^T H sigma by first computing the gradient g = dl/domega of the loss l with respect to the weights omega, and then computing the gradient of the inner product g^T sigma which is a scalar, with respect to the weights again. This gradient-over-gradient can be inner product multiplied again with sigma.

In order to save cost the authors suggest a drop out method setting a fraction of the terms of sigma to 0 which saves a fraction of the derivative computation and correspondingly drops terms of the trace sum.

The authors then present experimental results evaluating their suggested regularization method in combination with and compared to other regularization methods on
- CIFAR10 / Resnet18
- CIFAR100 / Wide Residual Network
- WikiText-2 / 2-layer LSTM

**Summary Of The Review:**

I do not recommend to accept the paper in its current form since the connection between the theoretical justification and the empirical evidence is weak and the experiments in the paper are not sufficient for an empirical understanding of how the suggested regularizer really works.

---

> ### Author Response · Authors · 2021-11-21
> **Author Response (Part 1)**
>
> Thank you for your suggestions.
>
> Above all, to make our method more convincing, we add new experiment on SEHT-H (maxIter=5). Our new result shows that using SEHT-H method separately can achieve 95.59 (+-0.06) test accuracy over 4 distinct runs on CIFAR-10. Our SEHT-D also achieve higher result when doing a new hyper-parameter search. With weight value 0.001, our SEHT-D (maxIter=1, prob=0.01) separately achieve 95.37(+-0.09) over 5 distinct runs. We also test SEHT-D with different maxIter {1, 5, 10}. SEHT-D (prob=0.01, maxIter=10) improve test accuracy 0.04 compared with SEHT-D (prob=0.01, maxIter=1). For efficiency and performance, SEHT-D (prob=0.01, maxIter=1) is good enough.
>
> We made an incorrect expression in our initial submission. We showed 95% confidence interval, while we are supposed to show the standard error. We have already corrected it.
>
> Here are our explanations toward your concerns.
>
> 1.	Q: “My feeling is that the connection between using the Hessian / Jacobian with respect to the logits vs with respect to the parameters is not as straight forward as it is made out to be in the paper and could be explained better…”
>
> When we consider more complicated models, we did an expansion w.r.t the units of models. The generalization bound concerns the output units of a linear model. A DNN is consist of many layers, with each layer being viewed as a linear model (except the nonlinear activation functions). Actually, we take a look at each unit in a neural network, which can be represented by the parameters. And the Jacobian Matrix in our statement is dl/dw, where w is the trainable parameters. So, the Jacobian matrix represents the gradient information. The process of gradient descent is to find the local or global minimum of loss function, at which point the gradient is 0. If gradient is continuous w.r.t parameters and the gradient is close to 0, then the norm of the gradient is small near minimum. That's the meaning of our statement. We don’t care about intermediate variables like dl/df. Gradient descent makes sure that dl/dw is near 0. Otherwise, if there is still large gradient, it will keep going gradient descent.
>
> 2.	Q: “Similarly, an explanation of the difference between the trace of the loss Hessian w.r.t. to logits vs w.r.t to the weights would benefit the paper…”
>
> Actually, we only focus on dl/dw of the neural network in this work. There is no need to consider the derivatives of the Hessian trace w.r.t. logits.
>
> 3.	Q: “is there a connection between the idea that increasing instability helps generalization to the literature that says that flat optima help generalization?”
>
> What we mean about instability is not the instability about the whole dynamical system. Also, our regularization method does not aim at changing the stability of the dynamical system. The purpose of our method is not to increase the instability of training process. The idea of stability is only about the equilibrium point, which is the final point of parameters we want to find by gradient descent. This kind of stability of the equilibrium point indicates a risk of over-fitting, because it's totally determined by the training data. Our method attempts to avoid choosing such point as our final result, which may increase the instability of the parameters. It will not change the dynamical system. It shares some similarity to Confidence Penalty, which prevents peaked distributions to make the system less confident on its prediction distributions. Our regularization method is to make the training process less confident to the equilibrium, which is supposed to be determined by training data. On the other hand, the notion of stability in dynamical system implies that it can be easily converged in the dynamical system. In our setting, the easily converged equilibrium point may not be the best one. The literature saying that flat optima help generalization is interesting, but currently we have no idea to find the connection between the instability of a dynamical system and the flat optima.
>
> 4.	Q: “There is no discussion of using the reverse-over-reverse autodiff method…”
>
> The specific implement of our SEHT-H and SEHT-D are computed by the autograd package of pytorch, which is a particular autodiff package. What we discussed about the stochastic estimators is a transformation of the variable to be calculated. The calculation of Hessian trace is very slow and requires a huge GPU memory. So, we define two stochastic estimators and we only have to calculate these two estimators. There are indeed derivatives in the process to calculate the estimators. We calculate the gradient by autograd package, and then we calculate the inner product of vector v and gradient. After that, we keep use autograd package to calculate Hv. Finally with an extra inner product, we obtain the estimators. The advantage of SEHT-D is to decrease the length of v and gradient, which decrease the cost in both two derivatives accordingly.

---

> > ### Author Response · Authors · 2021-11-21
> > **Author Response**
> >
> > 5.	Q: “There is currently no analysis of the trade-off of the bias of the trace estimator by using more dropout vs additional compute cost…”
> >
> > We search hyperparameters for our SEHT-H and SEHT-D. We tested prob value {0.01, 0.05}, maxIter {1, 5, 10}, weight value {0.1, 0.01, 0.001, 0.0001}. We will show the result with plots in the next version.
> >
> > 6.	Q: “There is also no analysis of how trace accuracy impacts generalization performance and trace itself…”
> >
> > The calculation of true Hessian trace is too difficult. For only one layer of resnet-18, it costs almost 2 minutes in one batch (size 32 of training examples). So, the real Hessian trace is extremely time-consuming. We can show our estimated hessian trace changes through training.
> >
> > 7.	Q: “It would be good to see the training and cross validation curves for different hyperparameters of overlayed penalty terms…”
> >
> > In the experiments on wiki-text dataset with LSTM and GRU model, we think we did a good hyper parameter search. However, in the experiments with CIFAR, we at first set the default parameters based on the experiments in relative papers. For instance, we set number of projections nproj = 1 and weight values λJR = 0.01 based on the experiments in the original Jacobian regularization paper. It's true we need to do more hyper-parameter tuning efforts. So, we are running new hyper-parameter search on all hyper-parameters including weight values. For example, for Label Smoothing and Confidence Penalty, we perform a grid search over weight values {0.001, 0.005, 0.01, 0.5, 0.1} on CIFAR-10. We find that 0.01 is best for Confidence Penalty and 0.1 is best for Label Smoothing. Confidence Penalty achieve 94.41 (+-0.02), Label Smoothing 94.26(+-0.13), which are lower than our newly tested result on SEHT-H 95.59 (+-0.06). We also search hyperparameter for our SEHT-H and SEHT-D. We tested prob value {0.01, 0.05}, maxIter {1, 5, 10}, weight value {0.1, 0.01, 0.001, 0.0001}. With weight value 0.001, our SEHT-D achieves 95.37(+-0.09), which is much higher than Confidence Penalty and Label Smoothing. We will also show the effect of choosing different maxIter and probability value in plot. We will keep on hyper-parameter search for the experiment on CIFAR-100. More results on hyper-parameter search will be added to the final version of this paper. We will also show the training and cross validation curves in plots.
> >
> > 8.	Q: “How does the added regularizer impact the training dynamics, e.g., faster or slower convergence in the beginning vs end of training and why?”
> >
> > In the CIFAR-10 experiment, the baseline needs 75.8 epochs on average to achieve test accuracy of 90%, while our SEHT-D (prob=0.01) only needs 61.8 epochs. Our newly tested SEHT-H (maxIter=5) only need 20 epochs on average. We will draw a plot to show the difference of convergence.
> >
> > 9.	Q: “State the generalization bound as a theorem as in the original paper (Theorem 4.1)”
> >
> > That’s a good idea. We will state it as a theorem in the revised version.
> >
> > 10.	Q: “In related work perhaps also connect to other works analyzing Hessian and Fisher matrix spectrum …”
> >
> > We will cite this relative paper in the revised version.
> >
> > All the added experiment results will be shown in the final version of our paper.  We hope that the additional experiments can demonstrate this a convincing paper.

---

> > > ### Author Response · Authors · 2021-11-23
> > > **Author Response (Part 3)**
> > >
> > > Our newly tested result on CIFAR-10 is below.
> > >
> > > |Method|Test Accuracy|
> > > |-|-|
> > > |Baseline with Weight-Decay|94.00 ($\pm$0.47)|
> > > |Jacobian|89.23 ($\pm$1.02)|
> > > |DropBlock|89.23 ($\pm$0.44)|
> > > |Sankar’s Method for Full Network|88.05 ($\pm$0.22)|
> > > |Sankar’s Method for Middle Network|88.13 ($\pm$0.12)|
> > > |Confidence Penalty|94.40 ($\pm$0.02)|
> > > |Label Smoothing|94.26 ($\pm$0.13)|
> > > |cutout|94.02 ($\pm$0.22)|
> > > |mixup|95.39 ($\pm$0.13)|
> > > |SEHT-D(maxIter=1, prob=0.01)|95.37 ($\pm$0.09)|
> > > |SEHT-D(maxIter=10, prob=0.01)|95.42 ($\pm$0.11)|
> > > |SEHT-D(maxIter=10, prob=0.05)|95.49 ($\pm$0.06)||
> > > |SEHT-H(maxIter=5)|95.59 ($\pm$0.06)|

---

### Official Review · Reviewer_imd7 · 2021-11-01

**Correctness:** 3
**Technical Novelty And Significance:** 4
**Empirical Novelty And Significance:** 2
**Recommendation:** 8
**Confidence:** 3

**Main Review:**

The paper makes a strong case for the proposed regularization scheme: penalizing the trace of the Hessian.  The empirical results included in the paper seem to indicate that the proposed regularization scheme improves network performance (and generalizability).  The technical details are accessible, and it is easy to follow the mathematical steps presented in the paper that lead to the final "efficient" algorithm for estimating the trace of the Hessian.  Section 3.4 that describes the linear stability analysis cogently argues why minimizing the trace of the Hessian is one way to improve the generalizability of the network.

The paper also suffers from some shortcomings.  While reading the paper, I stumbled upon a number typos: misspellings, incorrect double quotes (very common in LaTeX users), etc.  These can be easily fixed.

A more serious issue with the paper is its use of the word "efficient," as in, "to efficiently implements [sic] the Hutchinson method."  I did not see any experiments that showcase how efficient the proposed method is.  I would have liked to see some runtime figures that drive home the point about the efficiency of the proposed approach.  Is there a way to remedy this?

The paper actually presents two methods for estimating the trace of the Hessian.  The first method does not use dropout,  whereas the second, more efficient, method uses dropout.  The paper contains no results using the first method.  It would be nice to know the performance gap between the first and the second method.  In many situations performance and efficiency are inversely related.  Does that hold here too?  Alternately, may be deep neural networks upend this trend and the second method posts both better performance and better runtimes.

**Summary Of The Paper:**

The paper develops a new regularization method for deep neural networks.  The proposed methods penalizes the trace of the Hessian.  The paper adapts Hutchinson method for estimating trace of a positive semi-definite matrix for the purposes of estimating trace of the Hessian.  This adaptation uses a dropout mechanism to efficiently compute trace of the Hessian.   The paper also studies the effects of minimizing the trace of the Hessian using linear dynamical systems theory, and shows that lowering the trace of Hessian diminishes the stability of the equilibrium points in the parameter (i.e., weight) space.  Thereby, reducing overfitting and improving generalizability of the network.  The paper concludes with a set of experiments on standard deep learning benchmarks---CIFAR-10, CIFAR-100, and WIKI-TEXT2---and in almost every case the proposed SEHT-D scheme achieves best results.


**Summary Of The Review:**

Overall the paper is well-written.  It proposes a new regularization scheme that seems to improve deep neural networks performance on CIFAR-10, CIFAR-100, and WIKI-TEXT2 datasets.  The proposed scheme improves performance for both convolutional and recurrent neural networks.  The paper uses linear dynamical systems theory to argue that minimizing the trace of the Hessian improves the generalizability of the network.  Lastly, the mathematical description for "efficiently" computing the trace of the Hessian is easy to follow.

My only issue with the paper is that it doesn't include any results to support the assertion that the algorithm for estimating trace of the Hessian is efficient.

---

> ### Author Response · Authors · 2021-11-21
> **Author Response**
>
> Thank you for your suggestions.
>
> Above all, to make our method more convincing, we add new experiment on SEHT-H (maxIter=5). Our new result shows that using SEHT-H method separately can achieve 95.59 (+-0.06) test accuracy over 4 distinct runs on CIFAR-10. Our SEHT-D also achieve higher result when doing a new hyper-parameter search. With weight value 0.001, our SEHT-D (maxIter=1, prob=0.01) separately achieve 95.37(+-0.09) over 5 distinct runs. While the best result under a new hyper-parameter search for Confidence Penalty is 94.40(+-0.02) and 94.26(+-0.13) for Label Smoothing. We also test SEHT-D with different maxIter {1, 5, 10}. SEHT-D (prob=0.01, maxIter=10) improve test accuracy 0.04 compared with SEHT-D (prob=0.01, maxIter=1). For efficiency and performance, SEHT-D (prob=0.01, maxIter=1) is good enough.
>
> We made an incorrect expression in our initial submission. We showed 95% confidence interval, while we are supposed to show the standard error. We have already corrected it.
>
> 1.	Q: “I stumbled upon a number typos: misspellings, incorrect double quotes…”
>
> We will correct the typos in the next version.
>
> 2.	Q: “A more serious issue with the paper is its use of the word ‘efficient’…”
>
> The theoretical error bound and empirical relative error of Hutchinson method is introduced in Haim Avron and Sivan Toledo's work, which shows that Hutchinson method is efficient. For our problem, it's difficult to examine the error of estimated hessian trace because the real hessian is too hard to calculate. In our test, to calculate Hessian trace of one layer of parameters in resnet-18 cost almost 2 minutes for one batch (size 32 of training examples). So, the calculation of whole Hessian trace cannot be done. However, SEHT-D can be computed in almost 2 minutes in an epoch on one GTX 1080.  We can show our estimated hessian trace changes through training. We will also show the training and cross validation curves in plots.
>
> 3.	Q: “The paper contains no results using the first method…”
>
> We add new experiment on SEHT-H. We get 95.59% (+-0.06) test accuracy on CIFAR-10 over 4 distinct runs, which is higher than all other method tested. SEHT-H do have better performance.
>
> All the added experiment results will be shown in the final version of our paper.

---

> > ### Author Response · Authors · 2021-11-23
> > **Author Response (Part 2)**
> >
> > Our newly tested result on CIFAR-10 is below.
> >
> > |Method|Test Accuracy|
> > |-|-|
> > |Baseline with Weight-Decay|94.00 ($\pm$0.47)|
> > |Jacobian|89.23 ($\pm$1.02)|
> > |DropBlock|89.23 ($\pm$0.44)|
> > |Sankar’s Method for Full Network|88.05 ($\pm$0.22)|
> > |Sankar’s Method for Middle Network|88.13 ($\pm$0.12)|
> > |Confidence Penalty|94.40 ($\pm$0.02)|
> > |Label Smoothing|94.26 ($\pm$0.13)|
> > |cutout|94.02 ($\pm$0.22)|
> > |mixup|95.39 ($\pm$0.13)|
> > |SEHT-D(maxIter=1, prob=0.01)|95.37 ($\pm$0.09)|
> > |SEHT-D(maxIter=10, prob=0.01)|95.42 ($\pm$0.11)|
> > |SEHT-D(maxIter=10, prob=0.05)|95.49 ($\pm$0.06)||
> > |SEHT-H(maxIter=5)|95.59 ($\pm$0.06)|

---

### Official Review · Reviewer_YAnV · 2021-11-02

**Correctness:** 2
**Technical Novelty And Significance:** 3
**Empirical Novelty And Significance:** 1
**Recommendation:** 3
**Confidence:** 4

**Main Review:**

Strengths:

* The motivation of using jacobian and hessian trace norm regularizations is drawn from existing theoretical work, it's quite a nice way of tying it together with the extensive literature on generalization bounds. One comment worth adding though is that (I believe) all of those bounds are unfortunately vacuous. But even in that light, I think the idea is natural/worth writing a paper about.
* The initial downsampling (corresponding to p_1) is a clever way to make this work. Otherwise, instead of the 1.2-1.5x slowdowns, I would have expected to have seen at the very least a 2x slowdown (for two more derivatives), and practically speaking more like a 10-50x slowdown given how slow hessian vector products seem to be in practice.

Weaknesses:

* Unfortunately, the "better" results have heavily overlapping confidence intervals. In the top pane, the 2 std confidence interval is +/- 0.47/ Combined with say the seht-d + label smoothing, it's clear that none of these results are significant. I would suggest that the authors aggressively drive down the sizes of these confidence intervals. In the bottom pane, the comparison between mixup and seht-d + mixup suffers from the same problem. In fact, from these experiments, the only reasonable convlusion is that "mixup works really well".

* For comparison's sake, I think the authors really need to try the full version of the regularizer (with multiple iterations and prob=1.0). It's obviously going to be much slower, but if that doesn't work then the existing seht-d then I think something is probably wrong i.e., if the base claim is that "hessian regularization works", it is much less convincing if you don't directly test that hypothesis and go directly to approximations. I'd even be interested in seeing how much variance is introduced by the trace estimator, the two rounds of downsampling etc.

* I think there needs to be a careful study of hyperparameters of the regularizer i.e., prob and regularization coefficient etc.

**Summary Of The Paper:**

In this paper, the authors propose an efficient regularizer for the trace of the Hessian. To circumvent the extra computation needed to do the naive Hutchinson estimator (which pretty much amounts to taking *three* derivatives of the loss), the authors subsample the weights matrices, and then a second time internal to each weights matrix). Experimental results on CIFAR-10 and CIFAR-100 are presented: the proposed regularizer helps, but unfortunately not in a statistically significant way.

**Summary Of The Review:**

The idea is interesting, the approximation is nice, but the experimental results are too weak/incomplete to make this a convincing paper.

---

> ### Author Response · Authors · 2021-11-21
> **Author Response**
>
> Thank you for your suggestions.
>
> Above all, to make our method convincing, we add new experiments. We tested our SEHT-H (maxiter=5) on CIFAR-10 and get 95.59% (+-0.06) test accuracy over 4 distinct runs, which is higher than 95.39% (+-0.07) of mixup. Our SEHT-D also achieve higher result when doing a new hyper-parameter search. With weight value 0.001, our SEHT-D (maxIter=1, prob=0.01) separately achieve 95.37(+-0.09) over 5 distinct runs. While the best result under a new hyper-parameter search for Confidence Penalty is 94.40(+-0.02) and 94.26(+-0.13) for Label Smoothing. We also test SEHT-D with different maxIter {1, 5, 10}. SEHT-D (prob=0.01, maxIter=10) improve test accuracy 0.04 compared with SEHT-D (prob=0.01, maxIter=1). For efficiency and performance, SEHT-D (prob=0.01, maxIter=1) is good enough.
>
>
> We made an incorrect expression in our initial submission. We showed 95% confidence interval, while we are supposed to show the standard error. We have already corrected it here.
>
> 1.	Q: “the "better" results have heavily overlapping confidence intervals…”
>
> To this question we have following remarks.
>
> (1) On average accuracy, our methods are higher than other regularization methods (e.g., confidence penalty, label smoothing).
>
> (2) On standard error, our methods are smaller than confidence penalty, and at least not worse than label smoothing.
>
> (3) There is no overlapping in our newly tested results.
>
> (4) Actually, we are doing image classification and language modeling, not hypothesis testing in statistics. “Testing the hypothesis is equivalent to checking whether the null value \theta_0 is in the confidence interval.” (All of Statistics, Chap.10.1) We don’t know the real distribution of the test accuracy. Averages and standard deviations of the test accuracies are reported in our paper, not the “true” confidence intervals; we made an incorrect expression about this in our initial submission, we have already corrected it by showing the average and standard error. Generally, one cannot just use hypothesis testing on test accuracy in classification problems. Even if we want test a null hypothesis H_0: \theta_0 = \theta_1, it’s not accomplished by checking overlap in confidence interval. In the simplest linear model with normal assumptions, we need to use Welch's t-test to check hypothesis H_0. Thus, we think it is questionable to use this incorrect confidence interval overlaps to judge whether the experimental result is good or bad.
>
> (5) The mixup method belongs to data augmentation, which should not compete with our regularization method. Combining mixup with our regularizer can achieve the best results.
>
> 2.	Q: “I think the authors really need to try the full version of the regularizer…”
>
> We add new experiments on the full version regularizer (with 5 iterations and prob=1.0). As mentioned above, SEHT-H achieves 95.59% (+-0.06) test accuracy over 4 distinct runs, much better than other competing methods including mixup 95.39% (+-0.07).
>
> 3.	Q: “I think there needs to be a careful study of hyper-parameters of the regularizer…”
>
> (1)	We think our hyper-parameters search on wikitext-2 experiment is good enough.
>
> (2)	We are running a new hyper-parameter search on CIFAR-10 and CIFAR-100, including weight value and specific hyper-parameters of each method. For example, for Label Smoothing and Confidence Penalty, we perform a grid search over weight values {0.001, 0.005, 0.01, 0.5, 0.1} on CIFAR-10. We find that 0.01 is best for Confidence Penalty and 0.1 is best for Label Smoothing. Confidence Penalty achieve 94.41 (+-0.02), Label Smoothing 94.26(+-0.13)), which are lower than our newly tested result on SEHT-H 95.59 (+-0.06). We also search hyper-parameter for our SEHT-H and SEHT-D. We tested prob value {0.01, 0.05}, maxIter {1, 5, 10}, weight value {0.1, 0.01, 0.001, 0.0001}. With weight value 0.001, our SEHT-D achieves 95.37(+-0.09) over 5 distinct runs, which is much higher than Confidence Penalty and Label Smoothing. We will also show the effect of choosing different maxIter and probability value in plots. We will keep on hyper-parameter search for the experiment on CIFAR-100. More results on hyper-parameter search will be added to the final version of this paper.
>
> All the added experiment results will be shown in the final version of our paper. We hope that the additional experiments can demonstrate this a convincing paper. We regret that the reviewer did not recognize the contribution of our paper in the initial review, and hope that the reviewer can consider the review decision carefully once again. It appears that the recommendation score of 3 is unfair to our submission.

---

> > ### Author Response · Authors · 2021-11-23
> > **Author Response (Part 2)**
> >
> > Our newly tested result on CIFAR-10 is below.
> >
> > |Method|Test Accuracy|
> > |-|-|
> > |Baseline with Weight-Decay|94.00 ($\pm$0.47)|
> > |Jacobian|89.23 ($\pm$1.02)|
> > |DropBlock|89.23 ($\pm$0.44)|
> > |Sankar’s Method for Full Network|88.05 ($\pm$0.22)|
> > |Sankar’s Method for Middle Network|88.13 ($\pm$0.12)|
> > |Confidence Penalty|94.40 ($\pm$0.02)|
> > |Label Smoothing|94.26 ($\pm$0.13)|
> > |cutout|94.02 ($\pm$0.22)|
> > |mixup|95.39 ($\pm$0.13)|
> > |SEHT-D(maxIter=1, prob=0.01)|95.37 ($\pm$0.09)|
> > |SEHT-D(maxIter=10, prob=0.01)|95.42 ($\pm$0.11)|
> > |SEHT-D(maxIter=10, prob=0.05)|95.49 ($\pm$0.06)||
> > |SEHT-H(maxIter=5)|95.59 ($\pm$0.06)|

---

### Official Review · Reviewer_GDik · 2021-11-02

**Correctness:** 4
**Technical Novelty And Significance:** 3
**Empirical Novelty And Significance:** 2
**Recommendation:** 5
**Confidence:** 4

**Main Review:**

The paper starts well, with clear and simple goal - to introduce a new regualrizer. It motivates it through an existing generalization bound on linear models. Although, this bound does not translate to DNN models, I think this as a motivation is clearly presented and makes sense.
The stochastic estimators of the Hessian through Hutchinson method are well known and used in many places in ML.

Equation 11 is somewhat redundant, since anyone familiar with how modern autodiff packages perform efficient Hessian-vector products  would know that they are just restating well known facts. For the least they should be citing Perlmutter's paper on fast Hessian-vector products, which basically define this.
In this whole section on the Hessian estimator, I'm very unclear what is the benefit of the proposed extension SETH-D over SETH-H? Theoretically, due to dropping blocks of the whole network it should be more efficient. However, practically the way that modern autodiff packages work, it would be a significantly challenge to actually compute Hessian-vector products such that they benefit from this. E.g. if you just zero out those parts of `sigma` and perform still perform "dense" products on line 5 and 6 of Algorithm 2 it won't make any difference. Later in the results the authors do cite that based on the probability that they pick `p` they gain different run times. I think the authors should provide more details on how in practice they actually implement SEHT-D in order to take advantage of the structure of \sigma? These details although might look like not so important, are probably the only reason why one would even use SEHT-D over SETH-H.

I have quite a bit of issue with the whole section 3.4. The authors attempt to motivate the trace trough stability analysis. First, that they expand the idealized ODE, for which there is now some literature which uses backward analysis to derive more accurate ODEs and in practice show case that the idealized version is not well capturing what happens with the dynamical system. Second, the main point at the end of this whole section, talking about stability of equilibrium points, is to argue that the goal of the method is to have a "more unstable dynamical system". Why would we want that - I don't know. The last paragraph presents an argument that since standard training is a function only of the training data, we would like it to be unstable in order to avoid overfitting. However, the authors do not present any evidence or even elude to why that last statement would be true and I'm quite doubtful if it will at all. We can make training unstable in many other ways (injecting significant amount of noise for instance) which do not lead to better generalization. As a result I would advice the authors to either significantly change, or just remove this whole section, as it currently serves no purpose.

Finally, on the results section I have a few major issue with some of the numbers I'm seeing, which make me a bit doubtful of the overall results.

The WRN-28-10 has a baseline with 73% top 1 accuracy. The defualt pytorch open-source results from https://github.com/meliketoy/wide-resnet.pytorch indicate a 80% accuracy (ZCA - no-dropout 5e-4 weight decay). Perhaps the authors are suing different type of preprocessing scheme, or there are other differences, but this should be cleared out as to why they get so much worse results with the same model.

Another issue I have with the presented results, is that the authors do not mention to have done any search over the regualrizer penalty for any of the competing methods (and for theirs they do only two values - 0.01 0.05). They must mention why this is done - if they pick the values from one of the papers, they should make that more clear and also should make it clear that problems considered are equivalent to those in the reference. For many regualrizers their optimal weight changes between different problems. This might or might not explain some of the degradation results we see for some of the other techniques.

Probably, the final experiment is best execute, with significant hyper parameter search and good ablations, which makes it the only one that makes me trust its results, and unsurprisingly there the benefits are significantly more marginal then in the others. However, it is unclear what is Validation and Test Perplexy exactly? Do you train and cross-validate the hyper parameter to select them, and then retrain on train + validation and measure Test, or do you just measure the performance on the Test set of the best cross-validated model? Please clarify.

**Summary Of The Paper:**

The authors propose a regularizer for training DNNs - specifically adding the trace of the Hessian w.r.t. the parameters of the model.
They present two efficient stochastic estimators of the trace, based on the Hutchinson method and their own extension to it, specific to NNs.
The authors also provide a motivation of the regularizer through a known generalization bound and through the perspective of dynamical systems stability.
In three experiments - on CIFAR10, CIFAR100 and Wiki-Text they present results suggesting that the proposed method is performing better on average than other techniques.


**Summary Of The Review:**

Good and simple idea. I have some issues with lacking implementation details of the method that the authors use in practice. One of the sections seems completely redundant to the narrative, unless the authors provide better explanation for their final statements there. Some of the experiments seem to have lower numbers than publicly known results and in general there is a lack of exploring hyper parameters for other regualrizers. The last experiment is probably the only one that is done really well.

---

> ### Author Response · Authors · 2021-11-21
> **Author Response (Part 1)**
>
> Thank you for your suggestions.
>
> Above all, to make our method convincing, we add new experiment on SEHT-H (maxIter=5). Our new result shows that using SEHT-H method separately can achieve 95.59 (+-0.06) test accuracy over 4 distinct runs on CIFAR-10, which is much higher than all other method tested (such as newly tested result 94.40(+-0.02) for Confidence Penalty, and 94.26(+-0.13) for Label Smoothing). Our SEHT-D also achieve higher result when doing a new hyper-parameter search. With weight value 0.001, our SEHT-D (maxIter=1, prob=0.01) separately achieve 95.37(+-0.09) over 5 distinct runs. We also test SEHT-D with different maxIter {1, 5, 10}. SEHT-D (prob=0.01, maxIter=10) improve test accuracy 0.04 compared with SEHT-D (prob=0.01, maxIter=1). For efficiency and performance, SEHT-D (prob=0.01, maxIter=1) is good enough.
>
> We made an incorrect expression in our initial submission. We showed 95% confidence interval, while we are supposed to show the standard error. We have already corrected it here.
>
> Here are our explanations toward your concerns.
>
> 1.	Q: “Equation 11 is somewhat redundant…”
>
> The equation 11 is trivial in mathematics, but it’s necessary to our algorithm. We construct algorithm strictly follow the steps in equation 11. We calculate the gradient by autograd package in pytorch, which is a particular form of autodiff packages. Then we calculate the inner product of vector v and gradient. After that, we keep use autograd package to calculate Hv. Finally with an extra inner product we obtain the estimators. Actually, we didn't create a new derivative method. The specific implement of our SEHT-H and SEHT-D are computed by autodiff. In addition, we tried to calculate the Hessian without stochastic estimation. For only one layer of resnet-18, it costs almost 2 minutes in one batch (size 32 of training examples). So, the real Hessian trace is extremely time-consuming. We have to use stochastic estimators.
>
> 2.	Q: “I'm very unclear what is the benefit of the proposed extension SETH-D over SETH-H…”
>
> SETH-D is much faster in training speed. SETH-D is a two-step algorithm. Firstly, we randomly select layers in neural network with probability p_1 and randomly select parameters in the selected layers with probability p_2. We only have to calculate the gradient of selected parameters and corresponding partial sum to Hessian diagonal elements. The process of randomly selecting parameters in the selected layers with probability p_2, while estimating Hessian can be calculated in one step by Hutchinson method with our newly designed probability distribution. The difference is that the number of arguments is much smaller when calculating the derivatives with SETH-D. That’s benefit of the proposed extension SETH-D over SETH-H.
>
> 3.	Q: “I have quite a bit of issue with the whole section 3.4…”
>
> It seems that we didn't explain section 3.4 clearly. What we mean about instability is not the instability about the whole system. Also, our regularization method can’t change the stability of the dynamical system. The purpose of our method is not to add instability onto the training process, nor the structure of the network. This notion of stability is only about the equilibrium point, which is the final point of parameters we want to find by gradient descent. At the final end of the equilibrium point, this kind of stability indicates over-fitting, because it's totally determined by the training data. Through increasing the instability of the parameters, our method attempts to avoid such equilibrium point as our final result. It will not change the dynamical system. In essence, this instability idea is similar to confidence penalty, which prevents peaked distributions to make the system less confident on the training data. Our regularization method is to make the training process less confident to the equilibrium which are supposed to be determined by training data. On the other hand, the definition of stability in dynamical system implies that the training procedure can be easily converged in the system. In our problem, the easily converged equilibrium point may be not the best one because of the potential risk on overfitting.

---

> > ### Author Response · Authors · 2021-11-21
> > **Author Response (Part 2)**
> >
> > 4.	Q: “First, that they expand the idealized ODE, for which there is now some literature which uses backward analysis to derive more accurate ODEs and in practice show case that the idealized version is not well capturing what happens with the dynamical system.”
> >
> > Dynamical system is a well-known aspect to intercept gradient descent. There is definitely difference between empirical results and theorical results. However, the theory of dynamical system is important in our research. It gives us the intuition to think about equilibrium properties in neural network. Our experiment results also show that this is a good entry point to promote the deep learning models. We don’t mean that gradient descent in neural network is exactly a dynamical system. We just take a perspective of dynamical system to rethink neural network. But more precise theorical study on it is needed.
> >
> > 5.	Q: “The WRN-28-10 has a baseline with 73% top 1 accuracy…”
> >
> > We tested this pytorch open-source code (mean-std no-dropout 5e-4 weight decay) before we did our experiment in order to guarantee a decent baseline result.
> >
> > (1) The batch size is different, with 32 for our experiments and 128 for the default PyTorch open-source code. When we tested the SEHT-D method with batch size 64 and batch size 128, the GPU memory footprint was over 8GB. Due to the limitation of our GPU, we set batch size 32 for all competing methods. The default setting of batch size is 128 in that PyTorch open-source code, which may achieve 80% accuracy.
> >
> > (2) We also ran the default PyTorch open-source code (batch size 128) on our machine, but the top 1 accuracy can only be 76%-77% with mean-std normalization.
> >
> > (3) We noticed that our baseline accuracy appears to be lower than the reported open-source result. So, we refer to other papers for investigating results of WRN-28-10 with batch size 32. For example, the cutout method also test WRN-28-10 with Nesterov momentum 0.9 (we use momentum 0.9). In their paper, the top 1 accuracy is 73.96% for WRN-28-10 with batch size 32, which is similar to 73.79% reported in Table 2 of our paper.
> >
> > 6.	Q: “the authors do not mention to have done any search over the regualrizer penalty for any of the competing methods…”
> >
> > We picked the hyper-parameter values from the original papers of competing methods at first due to the time limit, but now we understand the default hyper-parameters may not yield the best results in our settings. Hence, we add a new hyper-parameter search for all methods we tested. For example, for Label Smoothing and Confidence Penalty, we perform a grid search over weight values {0.001, 0.005, 0.01, 0.5, 0.1} on CIFAR-10. Results show that 0.1 performs best for Label smoothing but 0.01 is the best for Confidence Penalty. Confidence Penalty achieve 94.41 (+-0.02), Label Smoothing 94.26(+-0.13)), which are lower than our newly tested result on SEHT-D 95.37(+-0.09) and SEHT-H 95.59 (+-0.06). We also search other hyper-parameters for our SEHT-H and SEHT-D. We tested probability value in {0.01, 0.05}, maxIter in {1, 5, 10}, weight value in {0.1, 0.01, 0.001, 0.0001}. We will show the effect of choosing different maxIter and probability value in plots. We will keep on hyper-parameter search for the experiment on CIFAR-100. More results on hyper-parameter search will be added to the final version of this paper.
> >
> > 7.	Q: “it is unclear what is Validation and Test Perplexity exactly…”
> >
> > The dataset of wikitext-2 is separated into three parts: train set, validation set, and test set. Apparently, the validation perplexity is the perplexity obtained on validation set while the test perplexity is on test set. We use the train set to learn regularized neural networks and employ validation set to find the best initial learning rate. We decrease the learning rate by factor of 4 when the validation error saturates. We run the experiment with every hyperparameter setting on the whole train set for 5 distinct runs. Then we choose the hyperparameter with best result.
> >
> > All the added experiment results will be shown in the final version of our paper. We hope that the additional experiments can demonstrate this a convincing paper.

---

> > > ### Author Response · Authors · 2021-11-23
> > > **Author Response (Part 3)**
> > >
> > > Our newly tested result on CIFAR-10 is below.
> > >
> > > |Method|Test Accuracy|
> > > |-|-|
> > > |Baseline with Weight-Decay|94.00 ($\pm$0.47)|
> > > |Jacobian|89.23 ($\pm$1.02)|
> > > |DropBlock|89.23 ($\pm$0.44)|
> > > |Sankar’s Method for Full Network|88.05 ($\pm$0.22)|
> > > |Sankar’s Method for Middle Network|88.13 ($\pm$0.12)|
> > > |Confidence Penalty|94.40 ($\pm$0.02)|
> > > |Label Smoothing|94.26 ($\pm$0.13)|
> > > |cutout|94.02 ($\pm$0.22)|
> > > |mixup|95.39 ($\pm$0.13)|
> > > |SEHT-D(maxIter=1, prob=0.01)|95.37 ($\pm$0.09)|
> > > |SEHT-D(maxIter=10, prob=0.01)|95.42 ($\pm$0.11)|
> > > |SEHT-D(maxIter=10, prob=0.05)|95.49 ($\pm$0.06)||
> > > |SEHT-H(maxIter=5)|95.59 ($\pm$0.06)|

---

### Decision · Program_Chairs · 2022-01-20

**Decision:**

Reject

**Comment:**

This paper regularizes deep neural networks via the Hessian trace.  The algorithm is based on Hutchison’s method, further accelerated via dropout.  Connection to the linear stability of dynamical system is discussed.  The proposed regularization shows favorably in the experimental results.

The idea of the method is clear.  The paper’s writing needs a lot of improvement because there are a number of grammatical errors.  The major technical concerns include: a) the experimental results are still not convincing; b) the explanation of favoring instability in the dynamical system that resorts to overfitting prevention (reviewer GDik).  I’ve read the rebuttal, but remain unconvinced.